# Multidrug Combinations against SARS-CoV-2 Using GS-441524 or Ivermectin with Molnupiravir and/or Nirmatrelvir in Reconstituted Human Nasal Airway Epithelia

**DOI:** 10.3390/pharmaceutics16101262

**Published:** 2024-09-27

**Authors:** Denise Siegrist, Hulda R. Jonsdottir, Mendy Bouveret, Bernadett Boda, Samuel Constant, Olivier B. Engler

**Affiliations:** 1Spiez Laboratory, Federal Office for Civil Protection, 3700 Spiez, Switzerland; 2Department of BioMedical Research, University of Bern, 3008 Bern, Switzerland; 3Department of Rheumatology and Immunology, Inselspital University Hospital, 3010 Bern, Switzerland; 4Epithelix Sàrl, Plan-les-Ouates, 1228 Geneva, Switzerland

**Keywords:** SARS-CoV-2, antivirals, molnupiravir, nirmatrelvir, GS-441524, ivermectin, reconstituted human nasal epithelia, COVID-19

## Abstract

**Background.** The emergence, global spread, and persistence of SARS-CoV-2 resulted in an unprecedented need for effective antiviral drugs. Throughout the pandemic, various drug development and treatment strategies were adopted, including repurposing of antivirals designed for other viruses along with a multitude of other drugs with varying mechanisms of action (MoAs). Furthermore, multidrug treatment against COVID-19 is an ongoing topic and merits further investigation. **Method/Objectives.** We assessed the efficacy of multidrug treatment against SARS-CoV-2 in reconstituted human nasal epithelia, using combinations of molnupiravir and nirmatrelvir as a baseline, adding suboptimal concentrations of either GS-441524 or ivermectin, attempting to increase overall antiviral activity while lowering the overall therapeutic dose. **Results.** Nirmatrelvir combined with molnupiravir, GS-441524, or ivermectin at suboptimal concentrations show increased antiviral activity compared to single treatment. No triple combinations showed improved inhibition of SARS-CoV-2 replication beyond what was observed for double treatments. **Conclusions.** In general, we observed that the addition of a third compound is not beneficial for antiviral activity, while various double combinations exhibit increased antiviral activity over single treatment.

## 1. Introduction

The emergence of SARS-CoV-2 in 2019 and its subsequent global spread resulted in an unprecedented need for antivirals for treatment and prophylaxis. Due to the urgency of an active global pandemic, repurposing drugs already approved for use in humans or at least demonstrated as safe for use became a priority, since novel drug development typically takes several years [1,2].

Since SARS-CoV-2 is a single-stranded RNA virus, RNA-dependent RNA polymerase (RdRp) inhibitors, such as remdesivir, were ideal early antiviral candidates [3,4,5,6]. Additionally, protease inhibitors that interfere with the initial cleavage of the coronavirus polyprotein upon virus entry were found to be highly effective [7,8]. Other non-antiviral drugs were also tested for activity against SARS-CoV-2, among them barcitinib [9,10], dexamethasone [11,12], tocilizumab [13,14], metformin [15,16], and ivermectin [17,18]. In addition to testing single drugs for activity against SARS-CoV-2, combination treatments targeting different aspects of the viral life cycle, or the resulting disease, were tested [19,20,21,22]. Multidrug treatment can prevent the emergence of resistant mutants, provide synergistic therapeutic benefits, lower overall dosage, and reduce toxicity [23,24,25,26].

We have previously investigated the antiviral activity of molnupiravir, both alone and in combination with various other drugs, and found that certain combinations resulted in increased antiviral activity compared to single treatment in reconstituted human nasal epithelia [27]. To further investigate the potential of molnupiravir, a nucleoside analog that acts as a competitive substrate for the RdRp resulting in accumulations of errors in the viral genome and incomplete viral replication [28], to serve as a basis of a multidrug treatment, we combined it with nirmatrelvir, a 3C-like protease (3CLpro) inhibitor that targets the SARS-CoV-2 main protease [29], in reconstituted human nasal epithelium, as this combination has been shown to be highly inhibitory against SARS-CoV-2 [30]. Lastly, we attempted to determine if adding a third compound, either GS-441524, an adenosine analog and the main plasma metabolite of remdesivir [31], or ivermectin, an anti-parasitic believed to have various MoAs, including interfering with importin-mediated nuclear transport [32], would provide additional inhibition compared to double combinations, potentially allowing for a lower therapeutic dose of each single component without losing antiviral activity.

## 2. Materials and Methods

### 2.1. Cell Culture

Vero E6 cells overexpressing transmembrane protease, serine 2 (TMPRSS2) were obtained from the Centre for AIDS Reagents (National Institute for Biological Standards and Control [33], and maintained in Minimum Essential Media (MEM, Seraglob, Schaffhausen, Switzerland), supplemented with 10% fetal bovine serum (FBS, Merck Biochrom, Schaffhausen, Switzerland), 0.1 U penicillin, 0.1 mg/mL streptomycin, and 1 m/mL Geneticin (Life Technologies, Gibco, Thermo Fisher, Basel, Switzerland) at 37 °C, 5% CO_2_, and >85% relative humidity (RH). Human nasal epithelial cells were obtained from patients undergoing polypectomy under informed consent, in accord with the Declaration of Helsinki (Hong Kong amendment, 2013) and with appropriate ethical approvals. Primary pulmonary cells were isolated from the tissue as previously described [34] and passaged once (p1). Two separate pools of nasal epithelial cells from 14 individual donors were seeded on 6.5-mm Transwell^®^ inserts (3470, Corning Incorporated, Oxyphen, Wetzikon, Switzerland) in MucilAir™ culture medium (EP04MM, Epithelix Sàrl, Geneva, Switzerland). Once confluent, air–liquid interface (ALI) was established by aspirating media from the apical compartment and maintained for at least 28 days at 37 °C, 5% CO_2_, and >85% RH to allow for mucociliary differentiation. The average culture time post-ALI was 51 days.

### 2.2. Virus Propagation

SARS-CoV-2 (reference strain: BetaCoV/France/IDF0571/2020, EPI_ISL_406596) was acquired from EVAg (Emerging Viral Diseases Medical Faculty, Marseille, France) and propagated on Vero E6 cells overexpressing TMPRSS2 in MEM supplemented with 2% FBS and the previously mentioned antibiotics at 37 °C, 5% CO_2_, and >85% RH for 3 days until harvest. Cell supernatant was centrifuged at 500× *g* for 15 min, aliquoted, and frozen at −80 °C until use. Tissue culture infectious dose 50 (TCID_50_/mL) was determined by a limiting dilution assay using the same cells and the Spearman–Kärber method [35,36,37].

### 2.3. Toxicity of Drug Combinations

The toxicity of the drug combination tested in the current study was evaluated by assessing the release of lactate dehydrogenase (LDH, Cytotoxicity Detection kit, Roche, Merck, Darmstadt, Germany) from the epithelial layer at 48 and 72 h post-treatment while replenishing drug-containing basal cell culture media every 24 h. Transepithelial electrical resistance (TEER) was also monitored at 24, 48, and 72 h, using the EVOM3 Volt/Ohm Meter and the accompanying chopstick electrode (STX2-plus, World Precision Instruments, Friedberg, Germany). Total resistance values (Ω) were converted to TEER (Ω × cm^2^) by first subtracting 100 Ω for the resistance of the polyester membrane and then multiplying by 0.33 cm^2^. Ciliary beating frequency (CBF) was assessed at 72 h by capturing 256 images at 125 frames per second at room temperature (RT) and calculated with Cilia-X (Epithelix Sàrl, Geneva, Switzerland).

### 2.4. Antiviral Testing

Prior to infection, MucilAir™ reconstituted nasal epithelium was washed twice with OptiMEM (Gibco, Thermo Fisher, Basel, Switzerland), warmed to 37 °C, and the basal media replenished with warm MucilAir™ cell culture media. An inoculum of 5 × 10^4^ TCID_50_ of SARS-CoV-2 diluted in OptiMEM were added to the apical compartment and incubated at 37 °C, 5% CO_2_, and >85% RH for 1 h. Mock controls were exposed to the same volume of OptiMEM-only for 1 h. Subsequently, the virus inoculum was removed, and the epithelium transferred to cell culture media containing molnupiravir (PubChem CID: 145996610); nirmatrelvir (Pubchem CID: 155903259); GS-441524 (PubChem CID: 44468216. Provided by California Institute for Biomedical research (Calibr) and WuXi AppTec China, Global Health Drug Discovery Institute (GHDDI); ivermectin (PubChem Reference Collection SID: 481107789) kindly provided by Prof. Dr. Johan Neyts University of Leuven, Belgium, alone or in combination. 5 μM remdesivir was applied as a positive control (PubChem CID: 121304016, Gilead Sciences Inc., Foster City, California, USA or MedChemExpress, (HY-104077, LucernaChem AG, Lucerne, Switzerland)). Apical virus release was assessed at 48 and 72 h post-infection (hpi) by washing the apical side with 200 µL OptiMEM for 10 min at 37 °C. During harvest of viral particles from the apical compartment, while the epithelium was submerged, epithelial integrity was also assessed by quantifying the TEER at 48 and 72 hpi. Test and control compounds were replenished every 24 h throughout the duration of the experiment. All compounds were diluted in DMSO, and final concentration of DMSO in the antiviral assay was standardized to 0.4%. All combination antiviral tests were performed twice in duplicate (n = 4). The following single concentrations were tested one to three times individually, with the following number of replicates: Molnupiravir 30, 10, 5, 2.5 µM (n = 3), 1 µM (n = 7); Nirmatrelvir 1, 0.5 µM (n = 3), 0.2 µM (n = 4), 0.1, 0.05 µM (n = 7); GS-441524 10, 5, 3 µM (n = 3), 1, 0.3 µM (n = 7); Ivermectin 10 µM (n = 7), 5 µM (n = 2), 2.5, 1 µM (n = 3), 0.25 µM (n = 6); and vehicle control (0.4% DMSO, n = 7). The detailed experimental layout has been described previously [27].

### 2.5. RNA Extraction and qRT-PCR

100 µL of apical wash was inactivated in 400 µL AVL buffer (Qiagen, Hombrechtikon, Switzerland) for 10 min at RT, after which 400 µL of absolute ethanol were added to each sample for complete viral inactivation, and samples were taken out of the BSL-3 laboratory for further processing. Viral RNA in apical secretions was extracted from 500 µL with the MagNaPure 96 system (Roche, Basel, Switzerland) according to the manufacturer’s instructions and eluted in 100 µL. SARS-CoV-2 RNA was quantified using the TaqMan™ Fast-Virus-1 Step master mix (Thermo Fisher, Basel, Switzerland) using the following cycling parameters, primers, and probe against SARS-CoV-2 non-structural protein 14 (*nsp14*): 50 °C for 1 min, 95 °C for 20 s, 45 cycles of 95 °C for 3 s and 60 °C for 30 s using the LightCycler 96 system (Roche, Basel, Switzerland); Fwd: 5′-TGGGGYTTTACRGGTAACCT-3′, Rev: 5′-AACRCGCTTAACAAAGCACTC-3′; probe: 5′-FAM-TAGTTGTGATGCWATCATGACTAG-TAMRA-3′ (protocol by Leo Poon, Daniel Chu, and Malik Peiris; School of Public Health, The University of Hong Kong, Hong Kong). Changes in gene expression were calculated using the 2^−ΔCt^ method and reported as fold reduction over virus control, 0.4% DMSO. At 72 hpi, epithelia were lysed in Qiazol (Qiagen, Hombrechtikon, Switzerland) for a total of 20 min at RT. Intracellular RNA was then extracted using the Qiagen RNeasy Plus Universal Mini kit according to the manufacturers’ instruction, eluted in 30 µL of nuclease-free water, and diluted 1:10 prior to gene expression analysis. Intracellular RNA was quantified using the SuperScript III Platinum One-step SYBR Green master mix (Thermo Fisher, Basel, Switzerland), the previously described primers against *nsp14*, and the following cycling parameters and primers against *GAPDH*: 50 °C for 1 min, 95 °C for 5 min, 50 cycles of 95 °C for 15 sec, 60 °C for 30 sec (53 °C for *nsp14*), 40 °C for 1 min followed by a melting curve to confirm product specificity; Fwd: 5′-GAAGGTGAAGGTCGGAGTCAAC-3′, Rev: 5′-CAGAGTTAAAAGCAGCCCTGGT-3′ [38]. Changes in intracellular gene expression were calculated using the 2-ΔΔCt method and reported as the fold reduction over virus control, 0.4% DMSO. Ct values > 40 were considered not-detected; this corresponds to a maximum log_10_ fold change of −7.

### 2.6. Determination of Infectious Titer

To quantify infectious virus in apical secretions, apical wash samples collected at 72 hpi were diluted in MEM supplemented with 2% FBS (Seraglob) and titrated in a limiting dilution assay using Vero E6 cells modified to constitutively express TMPRSS2 [31]. Cells were incubated for 3 days at 37 °C, 5% CO_2_, and >85% RH prior to cytopathic effect (CPE) determination by crystal violet staining. Infectious titer was determined as described above.

### 2.7. Statistical Analysis

Statistical significance between single, double, and triple treatments were determined by ordinary one-way analysis of variance (ANOVA) with Tukey’s multiple comparison test, and *p*-values < 0.05 were considered statistically significant. All analyses were performed with GraphPad Prism version 10.2.3 for Mac OS X, GraphPad Software, San Diego, California, USA, www.graphpad.com.

## 3. Results

### 3.1. Antiviral Activity of Molnupiravir and Nirmatrelvir, Alone or in Combination

A clear dose-dependent effect can be observed for both molnupiravir and nirmatrelvir when tested alone against SARS-CoV-2 in reconstituted human nasal epithelia. Lower concentrations of nirmatrelvir have a more pronounced effect, i.e., no detectable infectious titer at 72 hpi, after treatment with both 1 and 0.5 μM (Figure 1b, Table 1), while the same effect is only observed after treatment with 30 and 10 μM molnupiravir (Figure 1a, Table 1). Treatment with 1 μM molnupiravir and 0.1 μM nirmatrelvir result in similar residual infectious titers at 72 hpi, 5.88 and 5.83 Log_10_TCID_50_/mL, respectively. When combining these two concentrations, an additional statistically significant reduction of approximately 2.46 Log_10_TCID_50_/mL can be observed (Figure 1c, * *p* = 0.0122 and * *p* = 0.0148). Similar antiviral effect is observed after single treatment with 0.2 μM nirmatrelvir and by supplementing 1 μM molnupiravir. No antiviral effect is observed after treatment with 0.05 μM nirmatrelvir, either alone or in combination (Figure 1b,c, Table 2).

### 3.2. Antiviral Activity of GS-441524, Alone or in Combination with Molnupiravir and/or Nirmatrelvir

Antiviral activity of GS-441524 alone is pronounced for the three highest concentrations tested, i.e., 10, 5, and 3 μM, resulting in no detectable infectious titer at 72 hpi (Figure 2a). Treatment with 1 μM results in an average 0.73 Log_10_TCID_50_/mL reduction compared to non-treated vehicle control. When 1 μM GS-441524 is combined with 1 μM molnupiravir, a borderline statistically significant reduction of 2.4 Log_10_TCID_50_/mL is observed (Figure 2c, * *p* = 0.045). Supplementing nirmatrelvir, in any concentration, results in a highly significant reduction in infectious titer (0.2 μM *** *p* = 0.0002, 0.1 μM *** *p* = 0.0003, 0.05 μM ** *p* = 0.019; Figure 2c). Further adding 1 μM molnupiravir to combinations of GS-441524 and nirmatrelvir does not result in additional inhibition of viral activity.

### 3.3. Antiviral Activity of Ivermectin, Alone or in Combination with Molnupiravir and/or Nirmatrelvir

Only the highest concentration of ivermectin, 10 μM, results in antiviral activity at 72 hpi with a residual infectious titer of 3.88 Log_10_TCID_50_/mL (Figure 2b, Table 1). Combining 10 μM ivermectin with 0.2 and 0.1 μM nirmatrelvir results in borderline detectable infectious titer (0.98 Log_10_TCID_50_/mL, * *p* = 0.0057) and no detectable infectious titer (LLOD, * *p* = 0.0052), respectively. Large inter-assay variation is observed when the lowest concentration of nirmatrelvir is combined with ivermectin, suggesting borderline activity. Supplementing these double combinations with 1 μM molnupiravir does not result in statistically significant additional antiviral activity, while a non-significant reduction in infectious titer for the combination containing the lowest concentration of nirmatrelvir is observed in the triple combination (3.45 compared to 1.92 Log_10_TCID_50_/mL, Figure 2d, Table 2).

## 4. Discussion

In the current study, we attempted to optimize the antiviral activity of four compounds previously shown to inhibit SARS-CoV-2 in reconstituted human nasal airway epithelia by applying double or triple combination treatments, striving to reduce individual compound concentrations while maximizing antiviral activity.

Combination antiviral treatment offers a range of various benefits over monotreatment, including a reduced likelihood of escape mutants, possible synergistic effects, and less toxicity, by potentially reducing therapeutic doses. Combination treatments have historically been used for many RNA viruses, especially those viruses causing chronic disease and thereby requiring long-term treatment [24,39,40].

The emergence and rapid spread of SARS-CoV-2 in late 2019 resulted in an unprecedented need for antiviral drugs, and various pharmaceuticals with varying MoAs were tested against the virus and the resulting disease [41]. Additionally, various direct-acting antivirals were re-purposed after being designed for other RNA viruses, and SARS-CoV-2-specific antivirals were developed. Both inhibition of the RNA-dependent RNA polymerase (RdRp) with nucleoside analogs and prevention of the initial proteolytic cleavage of the polyprotein with protease inhibitors have proven successful against SARS-CoV-2 [42]. Intercepting these early steps of SARS-CoV-2 replication is relevant and applicable as both pre- and post-exposure prophylaxis and as early treatment. After the onset of symptoms, especially severe ones, treatment with direct-acting antivirals generally becomes less effective and more supportive, often immunomodulatory, treatment is required [43,44,45].

In the current study, we based the triple-drug treatment on combinations of molnupiravir and nirmatrelvir, a combination known to be a potent inhibitor of SARS-CoV-2 replication [46]. In reconstituted human nasal epithelia, a statistically significant reduction of infectious titer can be observed when 1 μM molnupiravir is combined with 0.1 μM nirmatrelvir, compared to single treatment, showing that additional antiviral effects can be observed by combining two suboptimal concentrations. However, adding 1μM molnupiravir to a higher concentration of nirmatrelvir (0.2 μM) does not add to antiviral activity compared to 0.2 μM nirmatrelvir alone and a double combination of 1 μM molnupiravir and a lower concentration of nirmatrelvir (0.05 μM) does not display a reduction in infectious titer compared to vehicle control. Addition of a third compound, either GS-441524 or ivermectin, to molnupiravir/nirmatrelvir combinations were well tolerated from a toxicity point of view but did not provide any additional antiviral effect.

However, double combinations of 1 μM GS-441524 with either molnupiravir or nirmatrelvir resulted in significantly lower residual infectious titers compared to single treatments. Interestingly, 1 μM molnupiravir and 1 μM GS-441524 result in about the same infectious titer at 72 hpi, 5.88 and 5.72 Log_10_TCID_50_/mL, respectively, but combinations of GS-441524 and nirmatrelvir perform better than combinations of molnupiravir and nirmatrelvir. Molnupiravir and GS-441524 are both nucleoside analogs inhibiting viral replication by causing lethal mutagenesis in new viral transcripts. Their metabolism is different, however, potentially explaining the differential responses when combined with nirmatrelvir in reconstituted nasal tissue. Additionally, the observed benefit of double treatments with nirmatrelvir and either molnupiravir or GS-441524 could be due to the presence of the SARS-CoV-2 3′-5′ exoribonuclease (nsp14), which is capable of excising incorrect nucleosides from the viral genome, thereby reducing the antiviral activity of nucleoside analogs [47].

A similar pattern of antiviral activity is observed for 10 μM ivermectin. However, the addition of 1μM molnupiravir does not provide an additional antiviral effect compared to ivermectin alone, while adding both 0.2 and 0.1 μM nirmatrelvir results in barely and no detectable infectious titer at 72 hpi, respectively. High inter-assay variability is observed for the lowest concentration of nirmatrelvir, indicating that the limit of antiviral activity of nirmatrelvir in combination with 10 µM ivermectin might lie between 0.1 and 0.05 μM. Although treatment with 10 μM ivermectin reduces residual infectious titer, both alone and in combination with nirmatrelvir, this concentration is impossible to achieve in human plasma, even with substantial overdosing of ivermectin [48,49]. The maximum observed plasma concentration of ivermectin, 0.25 μM, has no antiviral activity against SARS-CoV-2 in the reconstituted human nasal epithelia tested in the current study (Appendix A). However, this raises the question whether the oral bioavailability of ivermectin could be increased, thereby improving its systemic circulation and activity.

## 5. Conclusions

In conclusion, we were unable to elucidate additional direct antiviral effects against SARS-CoV-2 with triple combination treatment in reconstituted human nasal epithelia. However, combination treatment with two compounds at suboptimal concentrations did show additional efficacy beyond that observed for single treatments. The results presented in the current study further support the investigation of combination treatments against SARS-CoV-2 in more complicated infection models.

## Figures and Tables

**Figure 1 pharmaceutics-16-01262-f001:**
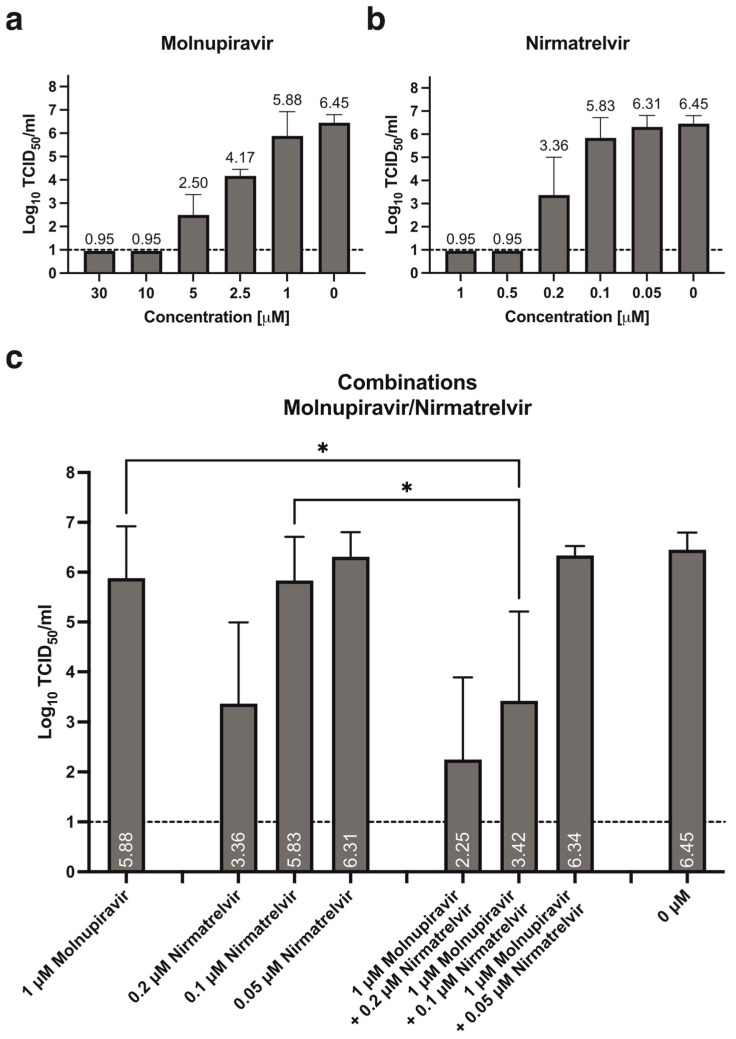
Treatment of reconstituted human nasal epithelia with molnupiravir and nirmatrelvir, alone or in combination. Dose-dependent antiviral effect of single treatment with (**a**) molnupiravir and (**b**) nirmatrelvir as assessed by residual infectious titer in apical secretions at 72 hpi. (**c**) Combination treatment with 1 μM molnupiravir and 0.2, 0.1, or 0.05 μM nirmatrelvir. Data presented as mean ± standard deviation (SD), n = 3–7 cultures, from 2–4 independent experiments. Statistical significance was assessed using a non-matched one-way analysis of variance (ANOVA) with Tukey’s multiple comparison test: 1 μM molnupiravir: * *p* = 0.0122; 0.1 μM nirmatrelvir: * *p* = 0.0148. Dashed line: lower limit of detection (LLOD), 1 Log_10_TCID_50_/mL.

**Figure 2 pharmaceutics-16-01262-f002:**
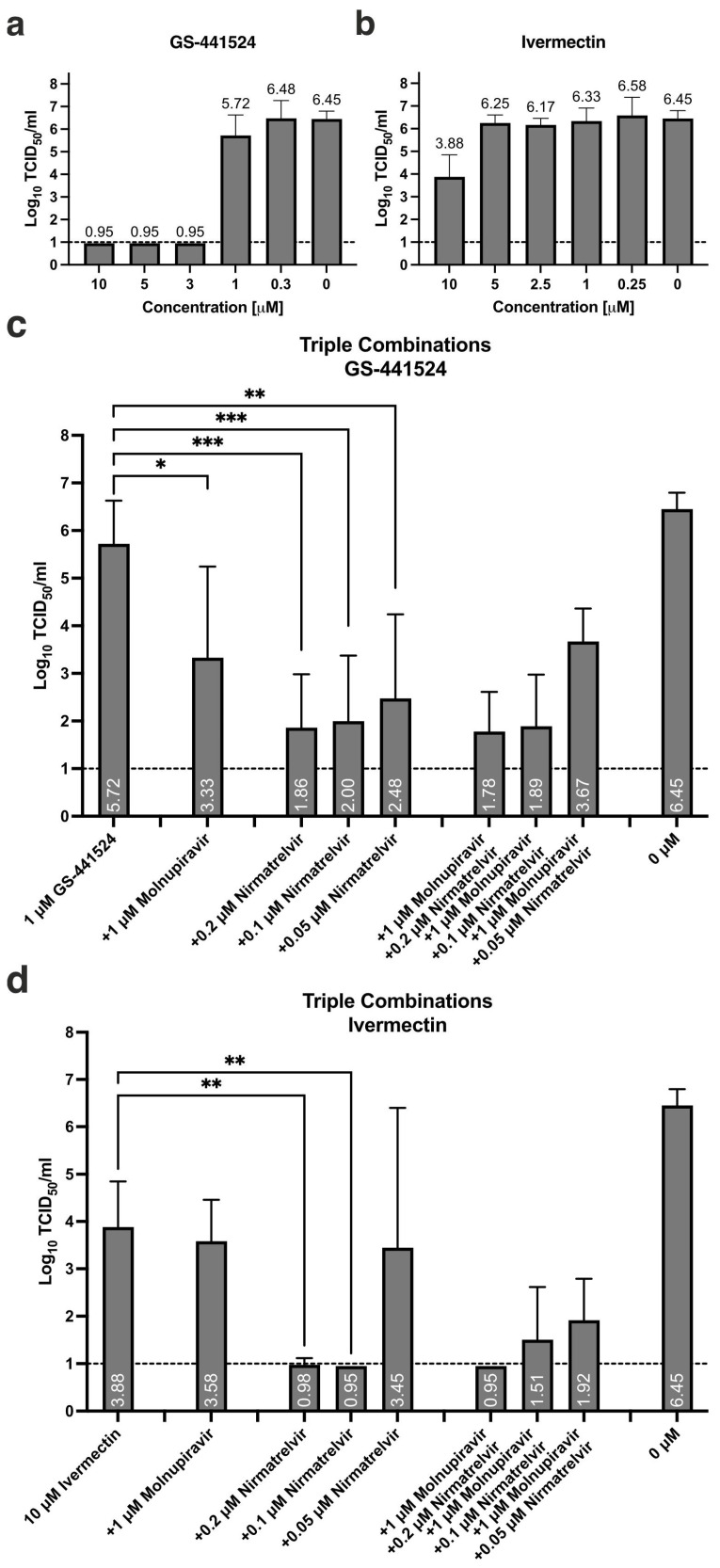
Triple combination treatment containing molnupiravir, nirmatrelvir, and either GS-441524 or ivermectin. Dose-dependent antiviral effect of single treatment with (**a**) GS-441524 and (**b**) ivermectin, as assessed by residual infectious titer in apical secretions at 72 hpi. (**c**) Triple combination treatment with molnupiravir/nirmatrelvir and GS-441524, * *p* = 0.0445, ** *p* = 0.0019, *** *p* <0.0003. (**d**) Triple combination treatment with molnupiravir/nirmatrelvir and ivermectin, ** *p* < 0.0057. Data presented as mean ± standard deviation (SD), n = 3–7 cultures, from 2–4 independent experiments. Statistical significance was assessed using a non-matched one-way analysis of variance (ANOVA) with Tukey’s multiple comparison test. Dashed line: lower limit of detection (LLOD), 1 Log_10_TCID_50_/mL.

**Table 1 pharmaceutics-16-01262-t001:** Summary of antiviral effects against SARS-CoV-2 after treatment with individual compounds. SD: standard deviation, IC: intracellular. n/a: not applicable, -: not titrated.

Concentration (µM)				Log_10_ Fold Change RNA						Infectious Titer	
Molnupiravir	PF-07321332	GS-441524	Ivermectin	48 h	±SD	72 h	±SD	IC	±SD	Log_10_ TCID_50_/mL	±SD
30				−3.69	0.33	−3.94	0.07	−4.62	0.04	LOD	n/a
10				−4.13	1.23	−3.20	0.43	−3.43	0.34	LOD	n/a
5				−2.02	0.38	−0.85	0.16	−0.83	0.26	2.50	0.71
2.5				−0.74	0.06	−0.47	0.02	−0.28	0.14	4.17	0.24
1				−0.21	0.10	−0.21	0.19	−0.23	0.20	5.88	0.96
0.5				−0.17	0.01	−0.07	0.07	−0.08	0.04	-	-
0.1				−0.32	0.06	−0.03	0.01	0.06	0.05	-	-
	5			−4.79	0.13	−4.17	0.64	−4.96	0.03	-	-
	1			−4.14	0.83	−3.99	0.16	−4.80	0.38	LOD	n/a
	0.5			−3.80	0.23	−4.10	0.18	−4.82	0.31	LOD	n/a
	0.2			−3.20	0.10	−2.97	0.67	−3.39	0.74	3.36	1.41
	0.1			−2.28	0.86	−0.93	0.60	−0.73	0.56	5.83	0.81
	0.05			−0.44	0.21	−0.19	0.12	0.00	0.16	6.31	0.46
	0.01			0.22	0.06	−0.13	0.26	−0.05	0.15	6.00	0.00
		10		−3.89	0.12	−3.85	0.74	−4.79	0.21	LOD	n/a
		5		−3.07	0.22	−4.07	0.27	−5.31	0.17	LOD	n/a
		3		−3.33	0.34	−3.38	0.33	−4.40	0.38	LOD	n/a
		1		−1.12	0.11	−0.88	0.10	−1.67	0.15	5.72	0.84
		0.3		−0.01	0.08	−0.16	0.12	−0.26	0.19	6.48	0.73
			10	−0.83	0.08	−0.55	0.17	−0.69	0.19	3.89	0.97
			5	−0.70	0.09	0.27	0.05	0.30	0.07	6.25	0.25
			2.5	−0.78	1.07	0.06	0.24	0.45	0.14	6.17	0.24
			1	−0.31	0.24	0.17	0.14	0.29	0.19	6.33	0.47
			0.5	−0.08	0.08	0.02	0.12	0.26	0.19	6.33	0.47
			0.25	0.16	0.20	0.04	0.08	−0.13	0.17	6.58	0.73
0	0	0	0	0.16	0.20	0.01	0.06	−0.01	0.06	6.45	0.32

**Table 2 pharmaceutics-16-01262-t002:** Summary of antiviral effects against SARS-CoV-2 after combination treatment. SD: standard deviation, IC: intracellular.

Concentration (µM)				Log_10_ Fold Change RNA						Infectious Titer	
Molnupiravir	PF-07321332	GS-441524	Ivermectin	48 h	±SD	72 h	±SD	IC	±SD	Log_10_ TCID_50_/mL	±SD
1				−0.21	0.10	−0.21	0.19	−0.23	0.20	5.88	0.96
	0.2			−3.20	0.10	−2.97	0.67	−3.39	0.74	3.36	1.41
	0.1			−2.28	0.86	−0.93	0.60	−0.73	0.56	5.83	0.81
	0.05			−0.44	0.21	−0.19	0.12	0.00	0.16	6.31	0.46
		1		−1.12	0.11	−0.88	0.10	−1.67	0.15	5.72	0.84
		0.3		−0.01	0.08	−0.16	0.12	−0.26	0.19	6.48	0.73
			10	−0.83	0.08	−0.55	0.17	−0.69	0.19	3.89	0.97
			0.25	0.16	0.20	0.04	0.08	−0.13	0.17	6.58	0.73
1	0.2			−2.97	0.12	−3.40	0.94	−3.90	0.79	2.25	1.64
1	0.1			−2.62	0.51	−2.41	1.47	−2.48	1.49	3.42	1.79
1	0.05			−0.98	0.13	−0.50	0.11	−0.18	0.29	6.33	0.19
1		1		−2.25	1.49	−2.09	1.26	−2.81	1.87	3.33	1.91
1		0.3		−1.74	0.93	−0.55	0.11	−0.95	0.38	5.83	0.54
1			10	−1.76	0.13	−2.23	0.40	−2.80	0.35	3.58	0.88
1			0.25	−0.35	0.31	−0.32	0.30	−0.55	0.14	6.00	0.33
	0.2	1		−3.11	0.19	−3.84	0.51	−4.58	0.18	1.86	1.12
	0.2	0.3		−3.07	0.26	−3.41	0.61	−4.08	0.65	2.31	1.57
	0.2		10	−3.03	0.24	−3.73	0.51	−4.67	0.41	0.98	0.14
	0.2		0.25	−2.83	0.37	−3.13	0.69	−3.63	0.81	2.92	1.55
	0.1	1		−3.09	0.26	−3.43	0.64	−4.42	0.29	2.00	1.37
	0.1	0.3		−2.46	0.31	−1.66	0.66	−1.85	0.43	4.83	0.27
	0.1		10	−3.06	0.18	−3.60	0.65	−4.13	0.75	LOD	n/a
	0.1		0.25	−2.73	0.40	−2.43	1.12	−2.29	0.77	4.42	1.34
	0.05	1		−2.79	0.85	−2.90	0.91	−3.53	1.47	2.48	1.76
	0.05	0.3		−1.95	1.25	−1.10	0.63	−1.59	0.74	5.83	0.27
	0.05		10	−2.22	0.43	−2.29	1.53	−2.70	1.46	3.45	2.95
	0.05		0.25	−0.65	0.09	−0.57	0.23	−0.44	0.20	6.50	0.27
1	0.2	1		−3.27	0.29	−3.70	0.48	−4.38	0.27	1.78	0.83
1	0.1	1		−3.29	0.24	−3.53	0.38	−4.30	0.12	1.89	1.08
1	0.05	1		−2.81	0.54	−2.59	1.08	−3.05	0.90	3.67	0.69
1	0.2	0.3		−2.95	0.15	−3.37	0.52	−3.89	0.12	1.70	1.24
1	0.1	0.3		−2.72	0.47	−2.55	1.11	−2.50	1.22	4.00	1.93
1	0.05	0.3		−1.27	0.36	−0.73	0.65	−1.20	0.68	5.50	0.86
1	0.2		10	−3.16	0.17	−3.71	0.49	−4.43	0.19	LOD	n/a
1	0.1		10	−3.14	0.25	−3.26	0.40	−3.64	0.51	1.51	1.11
1	0.05		10	−2.52	0.22	−3.07	0.45	−3.70	0.76	1.92	0.88
1	0.2		0.25	−2.89	0.33	−3.20	0.52	−3.55	0.43	3.08	0.69
1	0.1		0.25	−2.65	0.37	−2.60	1.03	−2.34	0.74	4.08	1.07
1	0.05		0.25	−1.15	0.22	−0.80	0.40	−0.87	0.19	5.42	0.57
0	0	0	0	0.16	0.20	0.01	0.06	−0.01	0.06	6.45	0.3

## Data Availability

The raw data supporting the conclusions of this article will be made available by the authors on request.

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
