# Peer review of "Multidrug Combinations against SARS-CoV-2 Using GS-441524 or Ivermectin with Molnupiravir and/or Nirmatrelvir in Reconstituted Human Nasal Airway Epithelia"

_pharmaceutics, 2024, doi:10.3390/pharmaceutics16101262_

Round 1

Reviewer 1 Report

Comments and Suggestions for Authors

Denise Siegrist et al. reported an interesting work about inhibitory activities of four compounds on SARS-CoV-2 in reconstituted human nasal airway epithelia model. The topic was attracting, and the content seemed to be innovative. This work might provoke discussion in the field of clinical medicine, drug discovery and public health. However, the reviewer was afraid that it was more suitable to be reconsidered by other biomedical journals held by MDPI Press, like International Journal of Molecular Sciences, Pharmaceuticals, Biomedicines and Molecules. Perhaps the editor and the authors could consider a transfer of the submission.

As a Volunteer Review Board member of several MDPI journals, the reviewer would like to explain the decision with more details. Of note, the journal Pharmaceutics is mainly focused on design, development, characterization and application of drug delivery systems and devices (microscopic and macroscopic ones). In some cases, it also publishes papers in biopharmaceutics. The current submission was not aimed to develop a drug delivery stem or device, neither to investigate the biopharmaceutical profile of the four compounds. Thus, with regret, this submission was probably out of scope of Pharmaceutics. On the contrary, the abovementioned alternative journals (International Journal of Molecular Sciences, Pharmaceuticals, Biomedicines and Molecules) can accept studies without emphasis on drug delivery.

Before transferred to another journal, the reviewer suggested to reckon the following simple recommendations to moderately improve the manuscript.

1.       The Abstract was a bit short (~130 words). Please consider to expand it to ~200 words to showcase more valuable information.

2.       The Introduction should not be written in only 1 paragraph. Please divide it into several paragraphs for better reading experience.

3.       The molecular structures of the four compounds should be displayed.

4.       Like #2, the discussion should be divided into several paragraphs.

5.       Please supplement an individual Conclusion Section.

6.       The format of pagination in the References was not unified. Please revise.

In summary, the reviewer hoped that the authors could make proper revision and transfer the manuscript to a more appropriate journal.

Reviewer 2 Report

Comments and Suggestions for Authors

This paper compared the antiviral activity of four compounds alone or in combination in reconstituted human nasal airway epithelia, on the basis that the combination antiviral treatment provides benefits over monotreatment, like reduced escape mutants, possible synergistic effects, and less toxicity. This study concentrated on the combinations of molnupiravir and nirmatrelvire. The results confirmed improved antiviral activity by combining 1µM molnupiravir and 0.1 µM nirmatrelvire (two suboptimal concentrations) than when used alone.  Concentration of nirmatrelvire was also an important factor, with a suggested optimal range of 0.05–0.1 µM. Also, ivermectin has a high threshold concentration (10 μM) which is impractical to achieve in plasma. Triple combination did not result in additional antiviral activity. The methods of this study are robust.  The results for useful for future antiviral medication development. This paper can be accepted after addressing the weaknesses below:

1.       Please specify how many times each test was repeated.

2.       Explanations to the observations should be provided. For instance, why “adding 1μM molnupiravir to a higher concentration of nirmatrelvir (0.2 μM) does not add to antiviral activity compared to 0.2 μM nirmatrelvir alone”. These two compounds have different acting mechanisms and work on different sites of the RNA. Explanations were also desired for other observations.  

3.       Line 169: spaces are needed at both sides of “=”  in “p=0.0122 and *p=0.0148”

4.       Figure 1: 1C should has consistent font size as 1A and 1B. 

Reviewer 3 Report

Comments and Suggestions for Authors

Peer review of Pharmaceutics 3158613:

Siegrist, D. et al.  “Multidrug combinations against SARS-CoV-2 using GS-441524 or ivermectin with molnupiravir and/or nirmatrelvir in reconstituted human nasal airway epithelia”.

This paper is about the outcomes of multidrug treatment for COVID-19 using host-targeting antivirals of molnupiravir, nirmatrelvir, and GS-441524 against SARS-CoV-2 in human nasal epithelial cells. Overall results showed that infectious virus titer is reduced with the combination of optimal concentrations of molnupiravir and nirmatrelvir in a single treatment. The addition of ivermectin to the molnupiravir-nirmatrelvir cocktail had low cellular toxicity but without additional protection against the virus. Also, ivermectin-molnupiravir and ivermectin-nirmatrelvir synergistic combinations lowered viral titers much more than single treatments. Other parameters, such as inter-assay variability, were observed and ultimately treatment with two antivirals showed efficacy over single drug usage.

Suggestions for scientific information:

1.      This manuscript is very well written and easily comprehended. To make the manuscript a bit more precise, the authors should consider including information regarding the host-targeting antivirals’ target protein(s) and resultant viral pathogenesis cellular entry and subsequent mechanism-of-action. For example, ivermectin docks to a specific virus spike receptor-binding domain ACE2 (Lehrer, S. and Rheinstein, Peter H. “Ivermectin Docks to the SARS-CoV-2 Spike Receptor-binding Domain Attached to ACE2”, In Vivo, 34(5), 3023-3026, 2020. DOI: 10.21873/invivo.12134). This information is provided for nirmatrelvir but information for remdesivir and molnupiravir should be included. The easiest place for this information to go would be in the “Introduction” or “Materials and Methods” sections.

2.      In “Conclusions”, it may be helpful to those who will read this paper, if the authors consider providing a sentence or two about how their results may affect in vivo testing.

Suggestions for English and grammar:

Lines 26-52: The right margin needs to be justified like the rest of the manuscript.

Line 42: Insert an “s” at the end of “combination” to make it plural.

Lines 60, 61, 69, 76, 99, 149: Correct term for relative humidity is “Rh”. / Change “CO2” to “CO2”.

Lines 63: Insert “)” after “2013” and before “,”.

Line 107: Change “ul” to “µl”.

Lines 164, 172, 202, 210: Capitalize “figure” and “table”.

Comments on the Quality of English Language

English is excellent.

Round 2

Reviewer 1 Report

Comments and Suggestions for Authors

The Reviewer appreciated the authors' work to improve the manuscript. The manuscript quality was again raised. However, the authors did not respond to that the paper might fell outside the scope of Pharmaceutics. I have no other questions.